# Evaluating Pedestrian Environment Using DeepLab Models Based on Street Walkability in Small and Medium-Sized Cities: Case Study in Gaoping, China

**Yibang Zhang** [1] **, Yukun Zou** [2] **, Zhenjun Zhu** [3] **, Xiucheng Guo** [1,* **and Xin Feng** [3]

1   School of Transportation, Southeast University, Nanjing 211189, China
2   Chengdu Institute of Planning and Design, Chengdu 610036, China
3   College of Automobile and Traffic Engineering, Nanjing Forestry University, Nanjing 210037, China
*   Correspondence: seuguo@163.com

**Abstract:** In small and medium-sized cities of China, walking plays an important role as a green and healthy way to travel. However, the intensification of motorized travel and poor planning of pedestrian transportation systems have resulted in poor travel experiences for residents. To encourage residents to change their mode of travel from motorized transport to greener modes, it is necessary to consider the characteristics of walking travel, design good walking street environments, and increase the advantages of walking in the downtown areas of small and medium-sized cities. In this study, a spatial environment model of a pedestrian street was constructed based on the walking score. Visual perception elements, street function elements, and walking scale elements were acquired by semantic segmentation of Baidu street view images obtained with the DeepLab model. Points of interest (POI) were obtained based on surveys, measurements, and the space syntax. Considering walking distances for small and medium-sized cities, the attenuation coefficient of a reasonable facility distance was adopted to modify the walking score. Based on the comprehensive score obtained, walking paths were divided into four categories: functionally preferred, visually preferred, scale preferred, and environmentally balanced. This categorization provides theoretical support for the design of pedestrian street space environments. Taking the pedestrian street in the city center of Gaoping in Shanxi Province, China as an example, the feasibility of the method and model was verified.

**Keywords:** small and medium-sized cities; street walkability; street space environment; walk score; DeepLab model

## 1. Introduction

As economic, political, and cultural development centers, the central parts of small and medium-sized cities play important roles in the development of these cities. By 2018, according to the *2019 Study on the Quality Development Index of Small and Medium-sized Cities in China*, there were 2809 small and medium-sized cities on the Chinese mainland, with 9.34 million km$^2$ of administrative regions, occupying 97.3% of the total land area. The population of these cities accounted for approximately 84.66% of the country's total population, and their aggregate economic output had reached 76.43 trillion yuan, which was approximately 84.89% of the total national economic output [1]. These figures demonstrate that the residents' income and development of small and medium-sized cities have contributed greatly to the social development and economy of China. Therefore, we should recognize the importance of small and medium-sized cities and consider them as key research subjects in urban structure optimization and new development.

Urban transportation planning is gradually shifting the goal to improving streets to facilitate walking and enhance urban sustainability [2,3]. Some large cities have already taken the lead (Shanghai Complete Street, Beijing Huoli Street, etc.). These measures

aim to improve street walking performance. The awareness of walkability in big cities continues to increase, and it is necessary for small and medium-sized cities to do some forward-looking research. Walking is a common choice for urban residents with regard to travel [4]. Walking not only plays an irreplaceable role in the urban transportation system, but also plays an important role in the sustainable development of cities [5]. Walking as a green transportation mode can alleviate traffic congestion and reduce greenhouse gas emissions [6]. However, there are certain problems with street pedestrian traffic in some small and medium-sized cities. For example, the walking facilities are not continuous, the walking environment is not friendly enough, and the walkability is unfriendly [7]. These problems not only affect residents' willingness to walk, but also increase the frequency of residents using motor vehicles to travel. Convenient pedestrian transportation facilities and a friendly walking environment can provide residents with higher destination accessibility for walking [8] and cause residents to change from motorized travel to walking.

The paper is organized as follows. Section 2 reviews the literature on walkability, street walking facilities, and the walking environment. Section 3 describes the relevant theories and methods of a street walkability study. Section 4 introduces the construction of the spatial environment model of a pedestrian street. Section 5 analyzes the spatial environment of Gaoping pedestrian street from the street functional elements, visual construction elements and walking scale elements. Section 6 describes the results of this study. Section 7 summarizes the study conclusions and suggests future areas of study.

## 2. Literature Review

Walking is a green and healthy lifestyle and can facilitate door-to-door services that public transportation cannot provide [9]. Increasing the proportion of people who walk can increase the vitality of cities, reduce the risk of chronic diseases, and extend people's lives [10]. Compared with travel by vehicles and bicycles, walking is an independent means of transportation that also occupies less road space per capita and does not consume energy directly. Walking has attracted significant attention in various fields, not only because it symbolizes a healthy lifestyle—that is, it has the characteristics of ecological and pollution-free travel—but also because it offers social benefits [11]. For example, walking environments allow people to conduct activities safely and conveniently, reach public facilities and gathering places quickly, and generate more social contact and productive trading opportunities [12]. Therefore, the excellent design of the walking environment can enhance the advantages of walking in small and medium-sized cities and encourage residents to shift from motorized travel to a greener travel mode.

Research on walkability mainly focuses on three aspects, namely, street walkability connotation, measurement of street walkability, and design of the walking environment. Street walkability, which is a description of spatial properties, is the ability of the social environment to guide travelers to choose to travel as pedestrians [13]. Southworth [14] observed that if you want to design a walkable city or community, you need to understand what walkability means in order to develop precise measures of walkability. He believes that walkability is the degree to which the built environment around the walkway is walkable, including the extent to which it provides a comfortable and safe environment for walking participants. Leslie et al. [15] defined walkability as the ability of the built environment and land use characteristics to attract residents of the area to walk for leisure, exercise, recreation, service, or work. In short, the greater the walkability is, the more people will choose to walk in an open urban environment. Moura et al. [16] agreed, arguing that the nature of walkability is the friendliness of the urban environment to pedestrians. Westerdijk et al. [17] studied the relationship between the street environment and the distance people were willing to walk in four European countries. The researchers divided the streets into seven levels according to a variable called "pleasure-level". After controlling for other variables, they found that when the street pleasure-level dropped by one level, the distance people were willing to walk decreased by 160 m or more. A study by Jaskiewicz et al. [18] demonstrated that simply measuring the distance to a destination

was not enough to measure walkability; the quality of the path was also significant. Many factors at the street level, such as the overall design of the street, visual attraction, facade transparency, landscape elements, lighting, and street activities play a positive role in encouraging walking [14]. Park et al. [19] proposed a route-oriented linear walkability concept named path walkability for use in micro-level walkability analyses that focuses on individual travelers and their routes.

In 2007, American researchers constructed a "walk score" evaluation based on the layout of street functional facilities, mainly considering the types and spatial layout of daily facilities. Factors such as the attenuation of dwelling density, street connectivity, land-use mix, flatness, walking distance, intersection density, and block length were introduced to quantitatively evaluate street vitality and improve the accuracy of the measurement [12–22]. The walk score is a widely accepted and widely used measure of walkability that has been widely used in many countries [23–25]. Without proper methods to measure walkability, proper walkability analyses cannot be carried out. Among the best methods are those that determine the values of walkability indexes such as the Global Walkability Index and the Asia-Index [26]. Ball et al. [27] and Owen et al. [28] established a multivariate analysis model for walkability that relates regional environmental factors to the choice of travel paths, based on residents' education level, physical condition, and travel purpose. They also summarized the difference in impact factors between commuting walking and leisure walking. With the help of a geographic information system (GIS)-based analysis platform, Manaugh [29] analyzed travel survey data for residents of Montreal, classified their trips by type, and found that there was a strong correlation between walkability and non-commuter travel. Duncan et al. [30] and Carr et al. [31] also used a GIS platform to verify that the walk score can represent a neighborhood community's support for walking. They selected various factors related to walking, such as distance attenuation and block size, to score types of street facilities and spatial layouts with respect to walking support. They found a significant positive correlation between the walk score and objective indicators such as connectivity, housing density, and public transport connectivity, as well as subjective indicators such as the total score of the physical activity environment.

The street is the most common public space in a city. Through the design of the walkway, the activity connection on both sides of the street is strengthened. The buildings and the road are successfully connected by the sharing of the area in front of the buildings with the pedestrian area of the road. Walkway designers strive to provide the best possible walking experience to encourage pedestrians to interact with the commercial interface and create a more dynamic business atmosphere. Goldberg [31–34] proposed the concept of the complete street, the goal of which is to provide a safe and accessible living street for travelers using all transportation modes. By the beginning of 2013, complete street policies had been implemented in 490 districts in 27 states in the United States. The main objectives of these policies included safety, environmental friendliness, and vitality. The concept of the complete street is embodied in the design of transportation networks, facilities, land use, and other aspects of urban environments, and has achieved positive results in practice in cities such as New York, San Francisco, London, and other megacities. The Pedestrian Environment Review System (PERS) was used by Transport for London to evaluate four walking paths from nearby metro stations to the ExCeL event venue. This evaluation focused on the capacity of pedestrian facilities when crowds gather, the readability of pedestrian orientation, the safety of pedestrians, and the quality of the walking environment. In view of the problems identified by the analysis of the evaluation results, more than 100 improvement measures that could produce results in the short term, such as optimization of intersections, strengthening of road information, and pavement re-paving, were proposed. Priorities for improvement projects were also identified in the improvement program. The differences in pedestrian street design methods for small and medium-sized cities in various studies are not clear.

The development level of public transportation in small and medium-sized cities is lower than that of large cities. Residents of these cities have a larger daily walking volume

and rely on walking to a higher degree. Improving the walking environment in small and medium-sized cities and promoting a healthy and civilized lifestyle has important research significance. The "Outline of the "Healthy China 2030" Plan" puts forward that health is an inevitable requirement to promote the all-around development of people. Lin et al. [35] found that walking is beneficial to reducing stress, and suggested that cities should provide more public spaces for walking exercise. As one of the most basic physical exercise methods to promote population health, walking is also an effective means to prevent and treat chronic diseases. For people who usually prefer walking, increasing walking during leisure time can reduce negative emotions [36,37]. Walk-friendly streets can not only promote the active walking behavior of the elderly, but also support the elderly to enhance their physical functions [38]. Residents in small and medium-sized cities are increasingly yearning for leisure activities, creating charming streets and improving residents' experience of the walking environment. These can attract more residents to walk as a form of exercise [36], and they can also attract more groups to travel for exercise.

Most previous studies on walkability have focused on the definition of walkability [39,40], the pedestrian street network structure [41,42], and the pedestrian space environment [43]. However, previous studies of pedestrian traffic systems have focused mostly on large cities or the centers of urban areas. Some studies have examined pedestrian travel behavior characteristics in small and medium-sized cities, but there are some gaps in the research to date on pedestrian traffic systems in small and medium-sized cities [44–46]. Concerning the morphological analysis of walking networks, previous studies have focused primarily on individual dimensions of static walking networks. These studies have provided great insight into the relationship between urban road network systems and urban pedestrian systems. There remain, however, some gaps in the research to date on the multi-dimensional analysis of pedestrian network structures. Quantitative analyses of walking space environments have typically relied on a walking index, which is not comprehensive enough to characterize the walking space based only on the street function. Furthermore, newly developed indices that reflect characteristics such as the abundance of trees and vegetation are mostly based on subjective observations, so it is difficult to collect data to quantify these characteristics objectively over a large area.

To fill these knowledge gaps, this study was conducted to establish an integrated approach to the assessment and design of pedestrian street environments in small and medium-sized cities. To address the limitation of the single measurement standard, a multi-factor characterization of pedestrian streets was developed that considers three aspects: street function, visual perception, and walking scale. In consideration of walk distance in the small and medium-sized cities, the facility attenuation factor coefficient will be improved to coordinate people's walking willingness. Furthermore, we will use the DeepLabv3+ model for a deep separation void convolution model for semantic segmentation of images and the bulk acquisition of urban elements. Taking the pedestrian street in the city center of Gaoping in Shanxi Province, China as an example, the feasibility of the method and model was verified.

## 3. Related Theories and Methods

### 3.1. Walking Distance Attenuation Effect in Street Functional Elements

Walking distance attenuation refers to the phenomenon that the mutual attraction between origin and destination tends to weaken with increasing walking distance for physiological and psychological reasons related to walkers. Thus, walking distance attenuation reflects the change of people's walking willingness with the travel distance, which means that people's willingness to walk decreases with the increase of walking distance generally. When daily travel is accomplished by walking, the attenuation coefficient of facility distance at the destination follows the law of walking distance attenuation.

Concerning walking attenuation, three factors are mainly considered: speed, time, and distance of walking, as comprehensively reflected in the walking distance attenuation. Considering the weight of the facility classification and the attenuation coefficient of the

walking radius, the walking attenuation distance can be divided into four parts, namely suitable walking distance, walking tolerance distance, walking resistance distance, and walking abandonment distance. People's suitable walking time is approximately 5 min. The average walking speed for pedestrians varies slightly according to age, as shown in Table 1. Considering all age levels, an appropriate walking distance is therefore within 500 m. The average walking speed for pedestrians varies slightly according to age, as shown in Table 1. Considering all age levels, an appropriate walking distance is therefore within 500 m [47]. According to the report entitled *Big Data Analysis on Public Transport in Major Chinese Cities in 2017*, released by Auto Navi Map in 2018 and based on investigations in 30 cities, as shown in Figure 1, the average walking distance in small and medium-sized cities is between 1000 m and 1300 m. Therefore, 1300 m was set as the walking distance tolerance boundary. In studying the walking landscape, it was found that a good walking environment can significantly increase pedestrian walking distance tolerance and even double it. However, the attenuation of the facility distance within this distance is significantly increased. Therefore, the range from 1300 m to 2400 m is called the walking resistance distance. A range of more than 2400 m is less likely to be walkable, so it is not taken into account and is defined as the walking abandonment distance.

**Table 1.** Average walking speed for walkers of different age groups.

| Age Group | Average Walking Speed (m/s) |
| --- | --- |
| 13–19 | 2.7 |
| 20–49 | 1.8 |
| 50–74 | 1.5 |
| >75 | 1.1 |

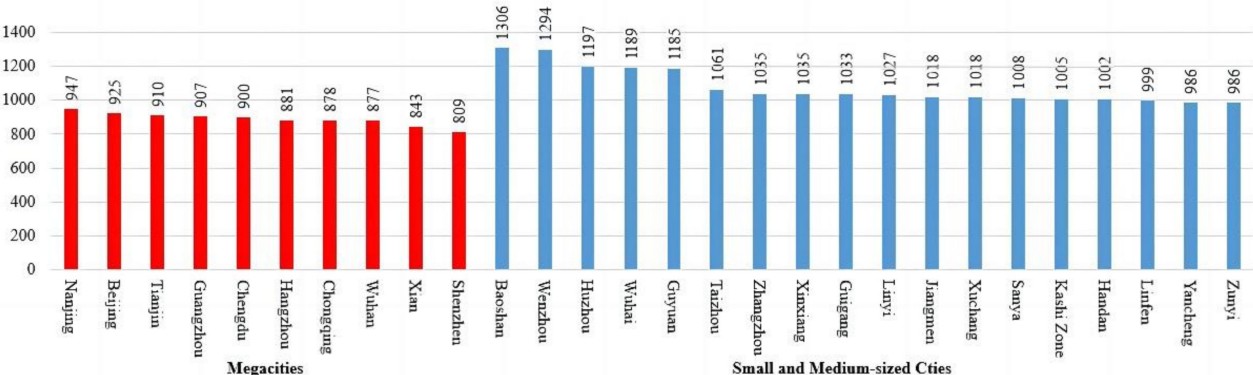

**Figure 1.** Average walking distance in each city.

### 3.2. Semanteme Division for Visual Construction Elements

Quantitative measurements of the visual appearance of urban environments have proven challenging because visual information is inherently fuzzy and semantically poor. To address this problem, this study relied on street images as agents of city appearance and used the newly developed image semantic segmentation technology to analyze city scenes in terms of scene elements [48]. DeepLabv3+, proposed by the Google team in 2018, was selected for the semantic segmentation of street pictures. This model is the latest optimized version of the DeepLab series of models, reflecting the state of the art [49]. Figure 2 shows the structure of this model.

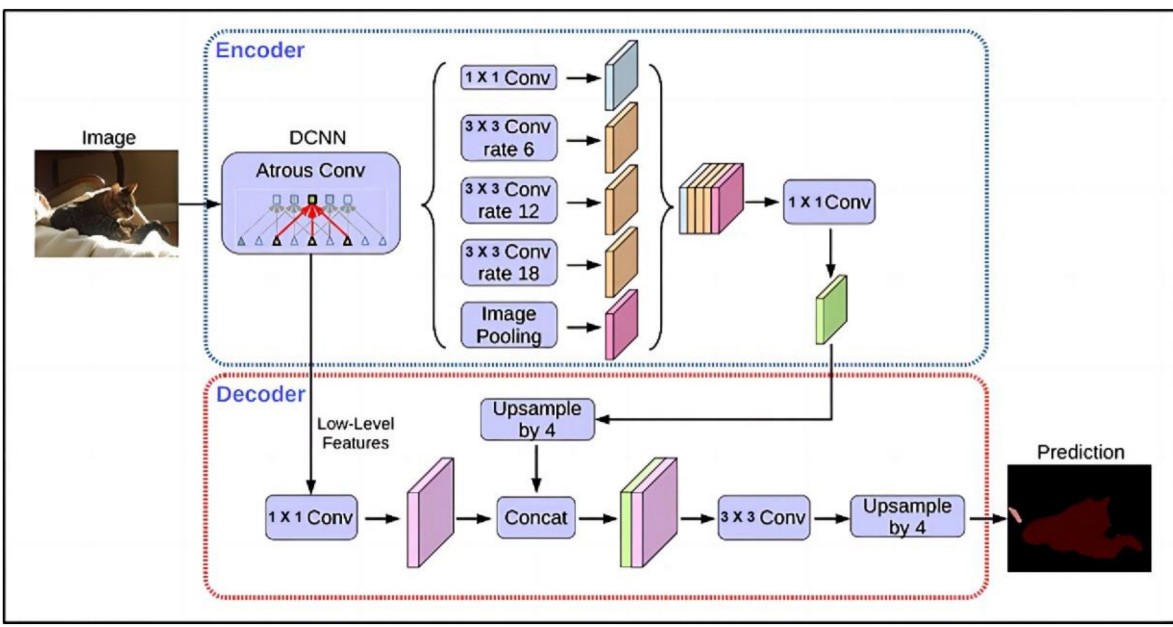

**Figure 2.** Structure of DeepLabv3+.

### 3.3. Overall Integration for Walking Scale Elements

The structure of the walking network in small and medium-sized cities has a closer internal function, which can attract people's choice of the walking path, and the movement of walking flow can affect the composition of the walking network. Space syntax can analyze the coupling effect between social logic and contradiction of the walking network in the development of small and medium-sized cities' walking network, and this coupling synergy can promote the multiplier effect of the walking economy and promote the central agglomeration of the walking network. The pedestrian network structure tends to cluster in a few routes in pursuit of lower economic cost of walking. The pedestrian network in small and medium-sized cities can reflect the agglomeration effect of the center. Therefore, for the change of pedestrian network, the agglomeration change of walking ability axis can reflect the change of walking network structure. In the interior of pedestrian street space in the central city of medium and small cities, the effect of pedestrian street network space on the main body of space use, that is, the pedestrian flow in different areas in the central city of medium and small cities, is more likely to be affected by the structure of the pedestrian street network itself when other spatial variables remain unchanged.

The overall integration degree of the pedestrian street represents the accessibility of the pedestrian street from the level of network topology [50]. Integration I represents the fabric relation space between the whole and the part of the system space by calculating the topological depth of each axis to all other axes. In the space, the obstacles between the fabric elements are small, and the integration is high if the agglomeration is high. Conversely, if the fabric elements are not closely related, the spatial structure is looser and the integration is relatively low. There are two key parameter variables in the calculation of integration, namely Relative Asymmetries ($RA_i$) and Real Relative Asymmetries ($RRA_i$) [51,52].

$$I = RA_i = \frac{2(MD_i - 1)}{n - 2} \tag{1}$$

$$MD = \frac{\sum\limits_{j=1}^{n} d_{ij}}{n - 1} \tag{2}$$

$$RRA_i = \frac{RA_i}{D_n} = \frac{(n-1)|MD - 1|}{n\left[\log_2\left(\left(\frac{(n+2)}{3}\right) - 1\right) + 1\right]} \tag{3}$$

where *MD* is the average depth value, the intermediate variable for calculating the integration, $d_{ij}$ is the topological distance between streets of the index *i* to index *j*, and n is the number of nodes in an abstract topological street network.

## 4. Methodology

Pedestrian street space is a complete system of interaction with walkers, and it is a comprehensive factor that affects residents' destinations and travel routes. In this study, the influencing factors of pedestrian street space are divided into three aspects: the choice of travel purpose, the perception of the travel environment, and the structure of travel space. Correspondingly, the index of the walking index model is divided into three categories: street function represents the travel purpose, travel environment represents the visual perception, and spatial structure represents the travel scale.

Based on the pedestrian index model, a spatial environment model of a pedestrian street (Figure 3), such as that shown in Equation (4) below, can be developed that considers the spatial layout of street function factors and the visual senses of spatial function factors for use in comprehensively analyzing the street space environment. Spatial function factors include sky openness, space color, greenery visibility, etc., which can be given weights and modified by walking distance attenuation and walking path width. Facilities can be divided into the following seven categories: commercial services, such as supermarkets, restaurants, barber shops, etc.; entertainment, such as a karaoke bar, teahouse, cinema, etc.; leisure, such as a park, or scenic spot; education, such as schools, training bases, and bookstores.; health care, such as hospitals, clinics, and pharmacies; administration, such as a police station, subdistrict office, or community center; and finance and telecommunication, such as a bank, ATM, or post office.

$$S = \sum_{i=1}^{n} \left( w_i \bullet \prod f(j) \right) \times \frac{100}{N} \tag{4}$$

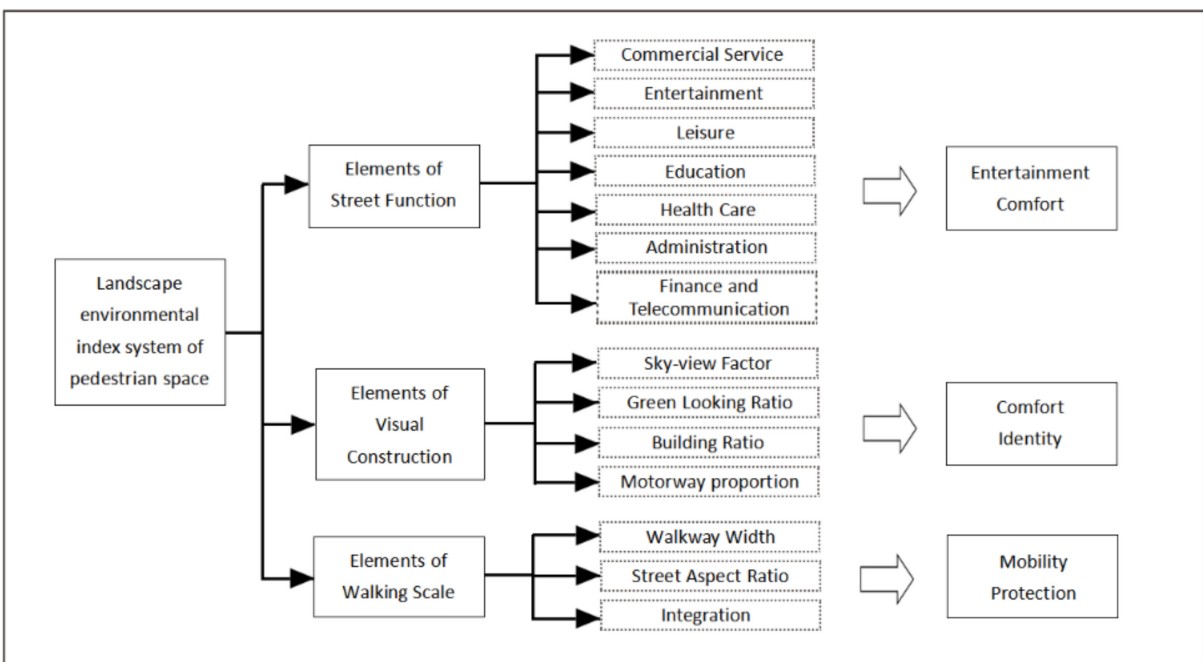

**Figure 3.** Spatial environment model of a pedestrian street.

In this equation, $w_i$ is the weight of the facilities' evaluation of index *i*, $f(j)$ is the attenuation factor's coefficient of index *j*, *S* is the pedestrian spatial environment score, and *N* is the total weight of the facilities' evaluation.

*S*, which ranges from 0 to 100, can be used to evaluate a pedestrian street at a holistic level. Visualization for each type of indicator can guide the design of the pedestrian street environment, as shown in Table 2 [53].

**Table 2.** The scale of the pedestrian street environment score.

| *S* | Standard |
| --- | --- |
| 0–24 | Very low walkability: unfriendly |
| 25–49 | Low walkability: few supporting facilities for walking |
| 50–69 | Medium walkability: some walkability facilities are within the walkability range |
| 70–89 | High walkability: meets basic daily walking needs |
| 90–100 | Very high walkability: supportive of daily walking trips |

*4.1. Pedestrian Street Functional Elements*

Pedestrian street functional elements, composed of facilities for walking, are an important part of the pedestrian landscape environment. The more diverse the functional elements are, the more pedestrian street vitality and pedestrian attraction increase, which makes a street more competitive in relation to optional paths. Differences between pedestrian street functional elements meet the needs of walking for different purposes and also lead to diversification of the needs of pedestrians for certain elements.

4.1.1. Classification Weighting

Lu [54] investigated pedestrian street functional elements in daily life and identified 20 common street functional elements, as well as their frequency of utilization, as shown in Figure 4.

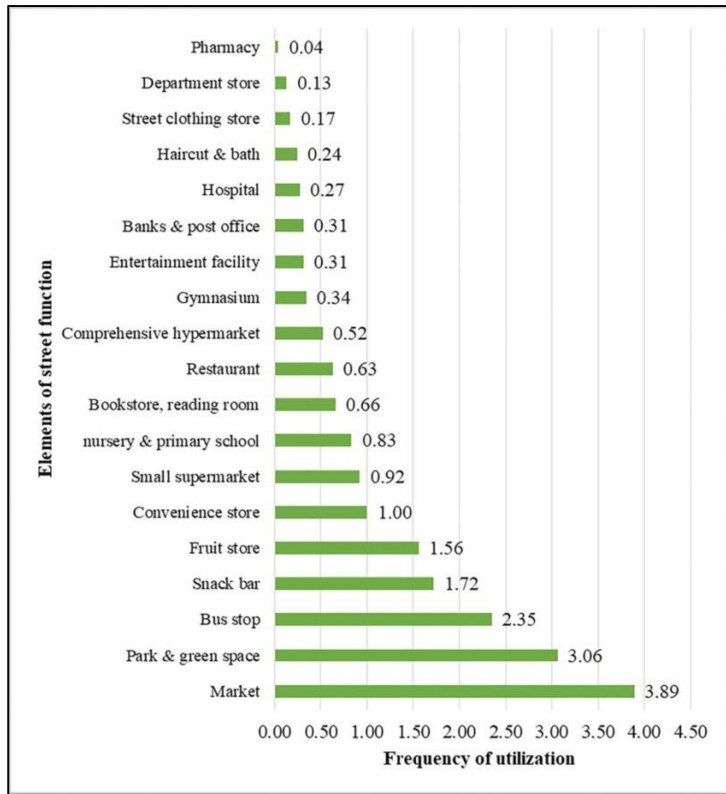

**Figure 4.** Frequency of utilization of street functional elements.

Based on the 20 classes, street functional elements were divided into seven categories and 22 subcategories according to the commonly used supporting functional facilities and travel purposes in the travel characteristics of residents in small and medium-sized cities, combining the characteristics and classification of small and medium-sized cities' street facilities. According to the report *"Evaluation of Walking Friendliness in Chinese Cities— Research on Street Function to Promote Walking"* released by the Natural Protection Association and School of Architecture and Tsinghua University, more detailed weight classifications need to be assigned to facility classifications.

According to the weight of different functional elements, street functional elements are divided into two categories by service type, as shown in Table 3. The first category is defined as homogeneous functional elements, which are the service facilities that are frequently used in daily life, such as convenience stores, pharmacies, communication business services, etc. The quality of service provided by these elements varies little, so people choose them according to the principle of proximity. The second category is defined as heterogeneous functional elements, whereby the quality of service of which is determined by the characteristics of the elements and differs greatly due to market competition. Their appeal is relatively insensitive to walking distance, and people are more likely to accept longer walking distances, within the affordable range, to access them.

**Table 3.** Pedestrian street functional element service features.

| Type of Functional Elements | Service Content | Element Types |
|---|---|---|
| Homogeneous functional elements | No significant difference in service quality | Financial posts and telecommunications, administrative management, leisure, medical and health (pharmacy, clinic) |
| Heterogenous functional elements | The quality of service is determined by the elements' properties | Business services, entertainment, health care (hospitals), education |

Taking into account the usage frequency and service characteristics, the pedestrian street functioznal elements were reclassified and weighted for the existing weighting results, as shown in Table 4. The weights were derived from *Walkability Evaluation of Chinese Cities*, issued by the Natural Resource Defense Council [55].

4.1.2. Distance Attenuation Rate of Walking Facilities

In different planning programs, the researchers for the distance decay function values and staging are diverse. In this study, based on the attenuation coefficient setting method in the report *Evaluation of Walking Friendliness in Chinese Cities –Research on Street Function to Promote Walking*, comparing the characteristics and scale of walking with those of large cities, the walking attenuation coefficient is optimized by combining the features of small-medium cities. The walking attenuation coefficient is set according to the attenuation distance of segments [55]. Within the suitable walking distance, the attraction of facilities to walking facilities will not weaken with the change of distance, and the distance attenuation coefficient is 1. Within the tolerance walking distance, an affordable range, the walking distance will be affected, and the distance attenuation coefficient is 0.88. Within the resistance walking distance, with the increase of the walking distance, the walking resistance emotion generated by pedestrians is greater and the distance attenuation coefficient is more than 0.25. If the walking distance is more than 2400 m, it will be deemed that walking facilities have no influence on walking, and the distance attenuation coefficient is 0. The specific walking attenuation function and coefficient are shown in Figure 5.

**Table 4.** Classification and weight distribution table of pedestrian street functional elements.

| Classification | | Weight | Total Weight |
|---|---|---|---|
| Commercial Service | Supermarket | 2 | 6 |
| | Restaurant | 3 | |
| | Barber Shop | 1 | |
| Entertainment | KTV | 1 | 4 |
| | Teahouse | 2 | |
| | Cinema | 1 | |
| Leisure | Park | 1 | 2 |
| | Scenic | 1 | |
| Education | School | 1 | 2 |
| | Training Base | 0.5 | |
| | Bookstore | 0.5 | |
| Health Care | Hospital | 3 | 6 |
| | Clinic | 2 | |
| | Pharmacy | 1 | |
| Administration | Police Station | 0.5 | 2.5 |
| | Subdistrict Office | 1 | |
| | Community Center | 1 | |
| Finance and Telecommunications | Bank | 1 | 2.5 |
| | ATM | 0.5 | |
| | Postal Communication | 1 | |
| Total | | 25 | 25 |

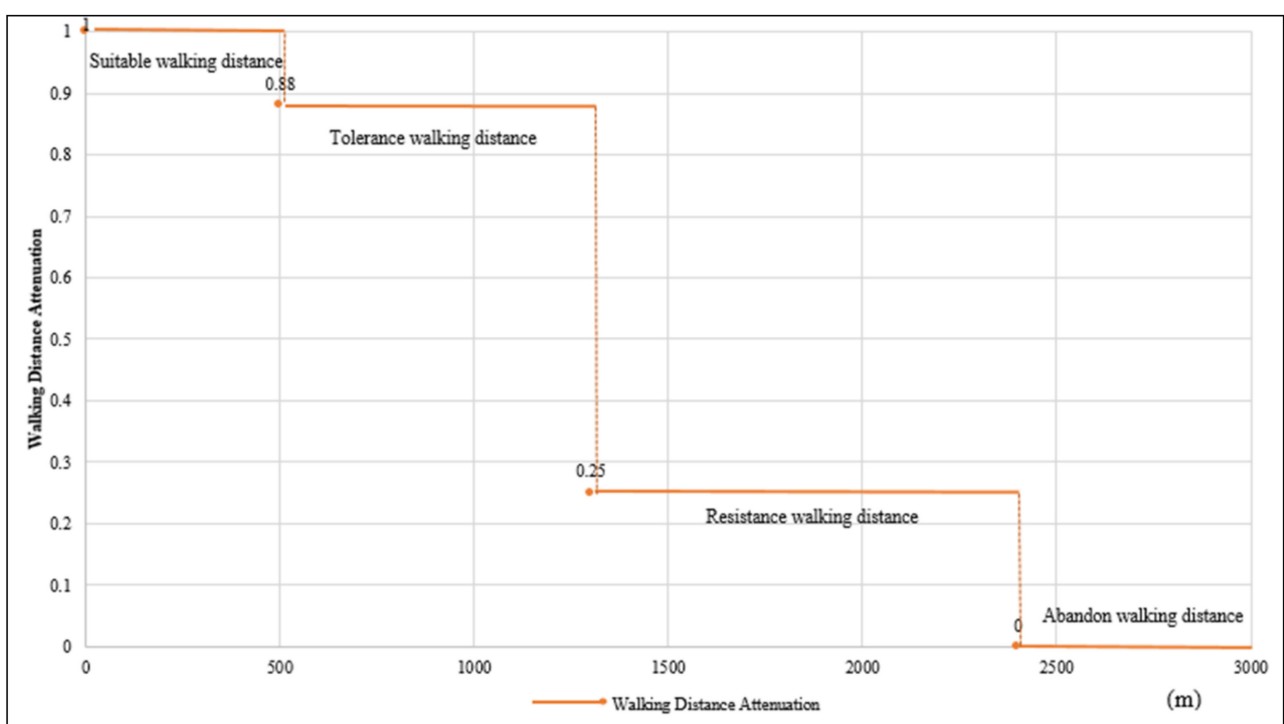

**Figure 5.** Distance attenuation rate of walking facilities at different walking distances.

*4.2. Visual Construction Elements*

In the process of implementation, Global Integration ($I_n$) is expressed by the inverse of $RRA_i$, which makes it easy to compare streets in the whole street spatial network system.

Supplementing existing walkability measures with data collected from street images using computer vision techniques is an efficient approach [56]. Daimler-Benz released the evaluation data set Cityscapes, which contains 5000 images of the high precision level annotation and level of more than 20,000 pieces of rough annotation of images. The images come from more than 50 different urban spaces in cities, street elements and backgrounds, and 30 physical elements' annotations including traffic signs, traffic actors, plants, and buildings. Based on commonality and importance to the street, the 19 categories' images were selected for instance segmentation during the test and were included in the calculation (Figure 6).

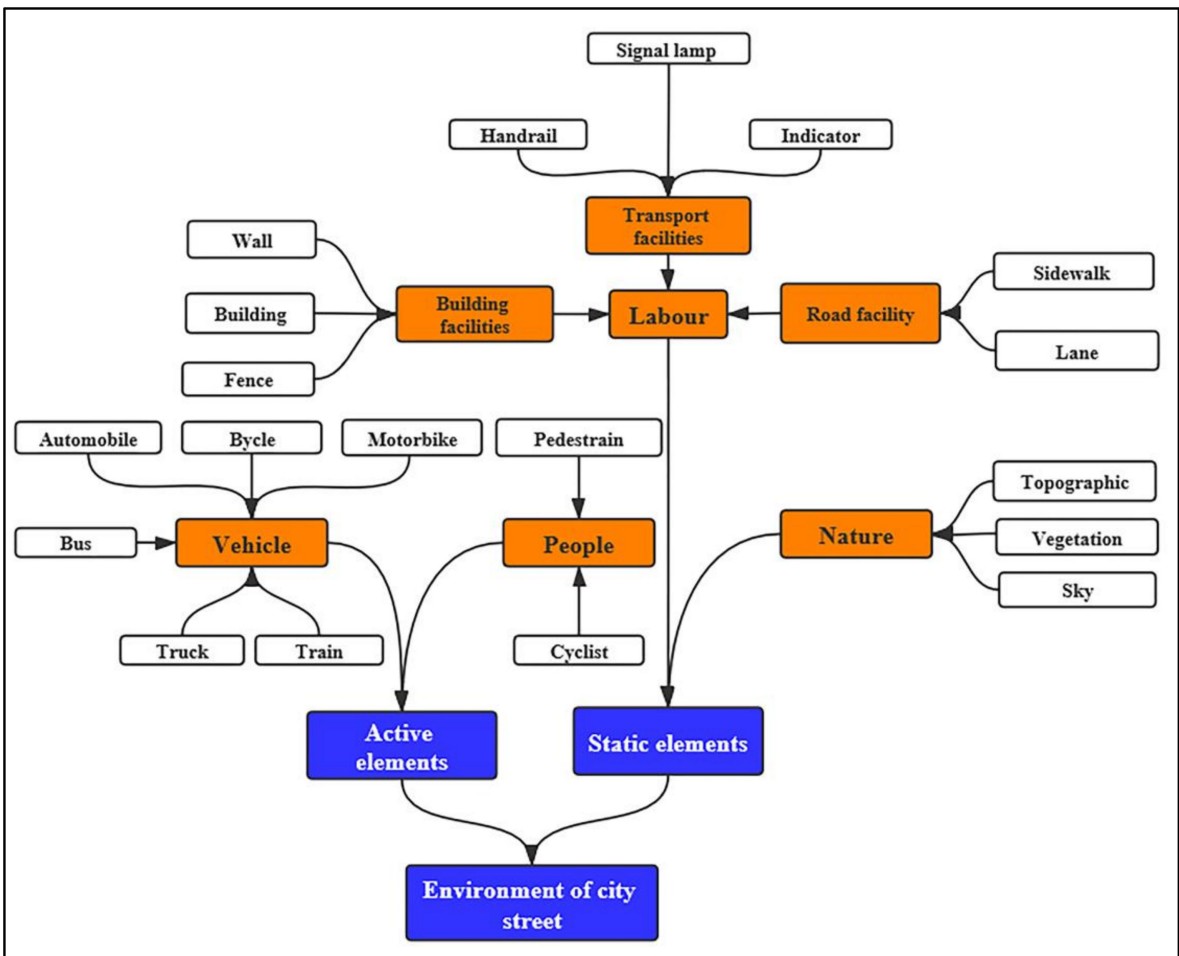

**Figure 6.** Cityscapes data set semantic tree.

*4.3. Walking Scale Elements*

The pedestrian scale element is a significant measurement criterion of street space and an important evaluation index of street spatial scale perception. In daily life, pedestrians' distance scales can roughly be divided into four categories, which are personal, length, social scale, and public scale. Among them, the social scale is considered to be the most appropriate scale for pedestrians to communicate in public spaces. In a space with a moderate scale or street interface, a reasonable walking space can create an appropriate social scale. In the space with this scale, pedestrians can feel more comfortable in the activity space, and are more likely to generate opportunities for communication, and extend communication time. On the contrary, pedestrians are more likely to have a sence of oppression and strangeness in wider scale space, which resulting in difficulty in gathering pedestrians.

### 4.3.1. Walkway Width

The width of the pedestrian path is usually measured on a horizontal scale. The activity range of walkers is within the red line of the road. From the perspective of the pedestrian space landscape, the width of the pedestrian path determines the activity behavior of the walkers and has a direct impact on the visual and spatial sense of the walkers.

The width of the walkway is influenced by street buildings and street furniture [57]. A wider pedestrian path is more conducive to the arrangement of more public facilities and the convenience of pedestrian activities. However, when the width of the pedestrian path exceeds a certain range, space will make the pedestrian lose the sense of place and reduce the pedestrian perception of street functions.

The width of the pedestrian path is an easily accessible indicator which can be obtained from road planning data, or, if available, from field surveys, such as the cross-sectional form of the road, the actual width of the footpath, etc.

### 4.3.2. Street Aspect Ratio

The whole street space is in the visual field of the walker, and the aspect ratio of the street is the walker's direct perception of the visual field. More specifically, the imbalance between the street width D and the height H of buildings along the street will affect the way walkers look at the street environment and the walkers' viewing range [58,59]. When the ratio of D/H is greater than 2, the vertical viewing angle of buildings within the visual range will be less than 26°. The wide space of buildings along the street will make walkers feel the increased sense of dispersion, and the functional atmosphere of the street will be blocked. When the ratio of D/H is less than 1, the top of the building will be more than 45° when viewed from an upward perspective. The spatial cohesion is enhanced and the horizontal perspective cannot include the panorama of the building. When the ratio of D/H is within the range, the harmonious proportion makes walkers feel that the walking street space is cohesive and stable without strong pressure.

### 4.3.3. Walking Scale Weight

According to the distribution law of overall integration value, pedestrian path width, and street aspect ratio, different values in the spatial environment evaluation model of pedestrian streets are given weights, as shown in Table 5 below [60].

**Table 5.** Walking scale weight.

| Index | Value Range | | |
|---|---|---|---|
| Overall integration | <0.8 | 0.8–1.5 | >1.5 |
| Walkway width | mixed traffic flow | Separate walkway but its width is not recommended | Separate walkway and its width recommended |
| D/H | >3 | (0, 1) or (2, 3) | (1, 2) |
| Weight | 1 | 3 | 5 |

## 5. Case Study

### 5.1. Study Area

Gaoping, located in the southeastern part of Shanxi Province (shown in Figure 7), has a permanent population of 490,000, an urbanization rate of 53%, and a GDP of 20 billion yuan. The urban built-up area is surrounded by mountains and rivers. Due to the relatively small scale of the city, it has maintained a good relationship with the surrounding hills, so it maintains a natural beauty. The Danhe River and the Xiaodongcang River also basically keep their original flow, which is conducive to creating an environment suitable for slow traffic.

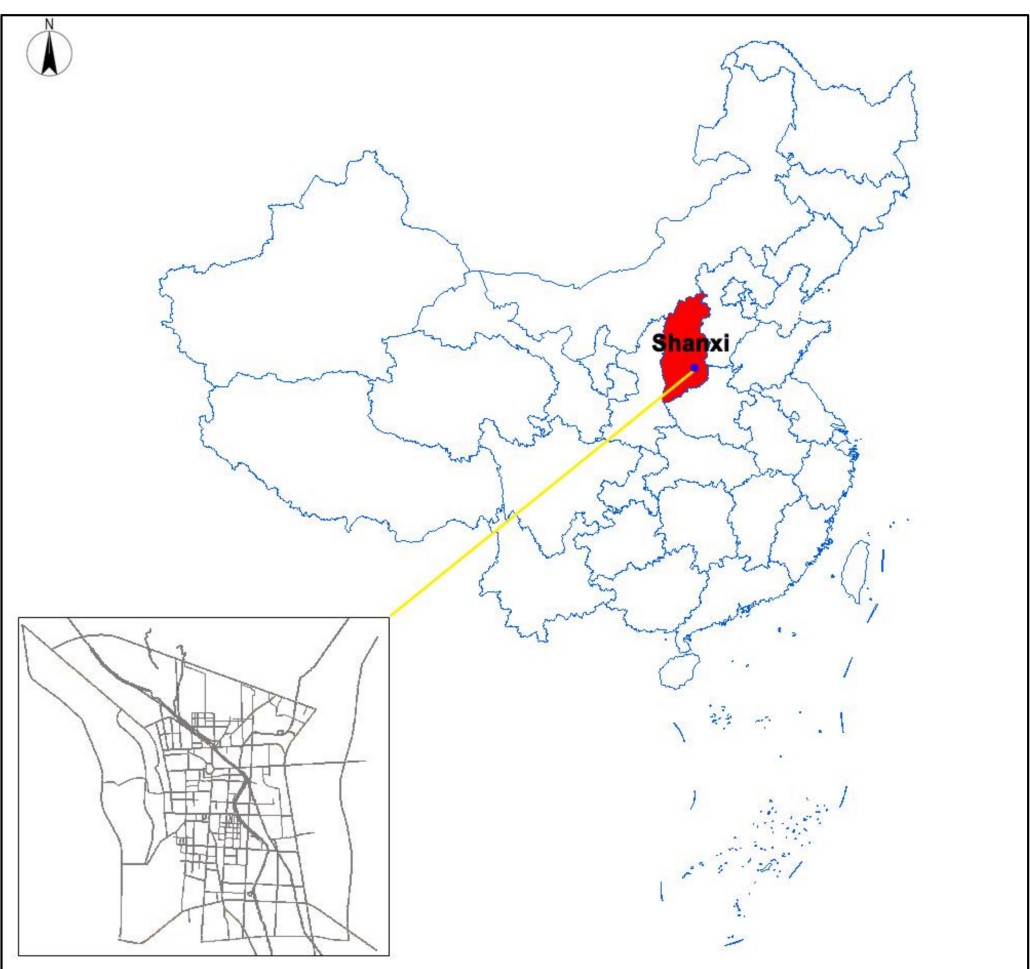

**Figure 7.** The geographic space pattern of Gaoping.

### 5.2. Data and Processing

Through POI, there were 1008 functional elements of streets in the downtown area of Gaoping, with a total of 20 categories, 1053 valid street view pictures, 269 supplementary street view pictures, and a 65-min street-view video and the cross-sections of 76 streets investigated.

### 5.2.1. Street Functional Elements

Among the 1008 street functional elements obtained, the two categories of commercial services and education accounted for more than 50%, and the distribution of various street functional elements was relatively concentrated, primarily on the axis of main pedestrian streets in the downtown area, which is shown in Figures 8 and 9.

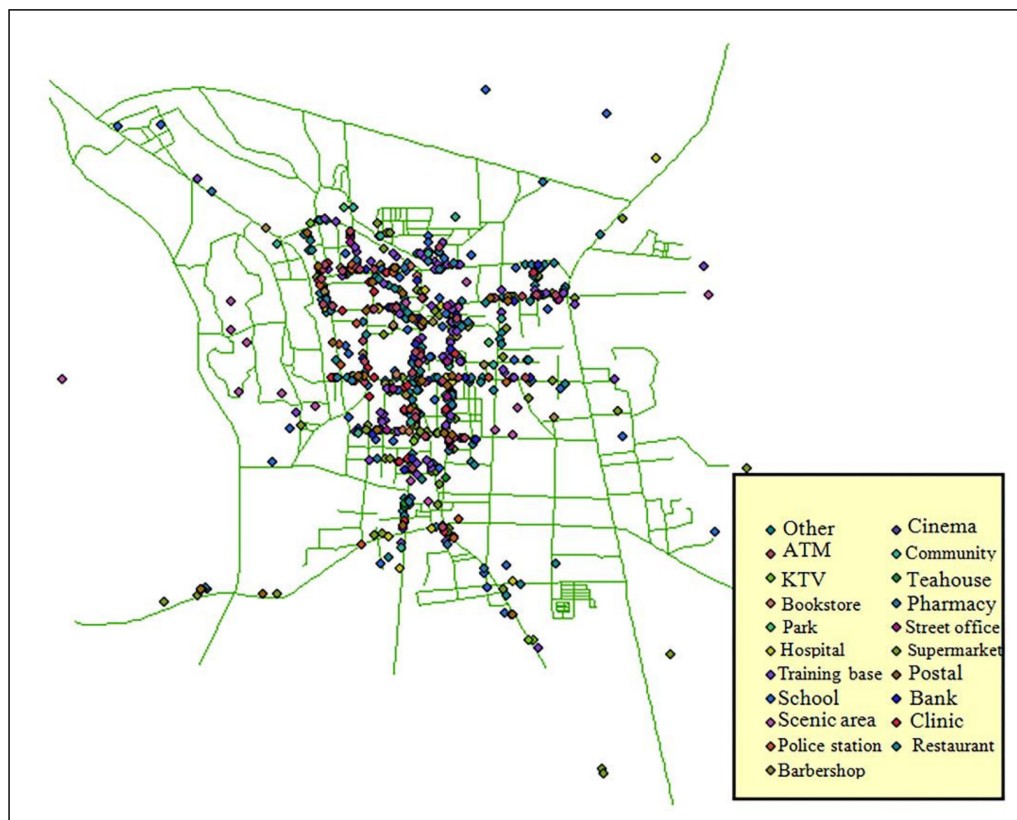

**Figure 8.** Distribution map of functional elements of downtown streets in Gaoping center.

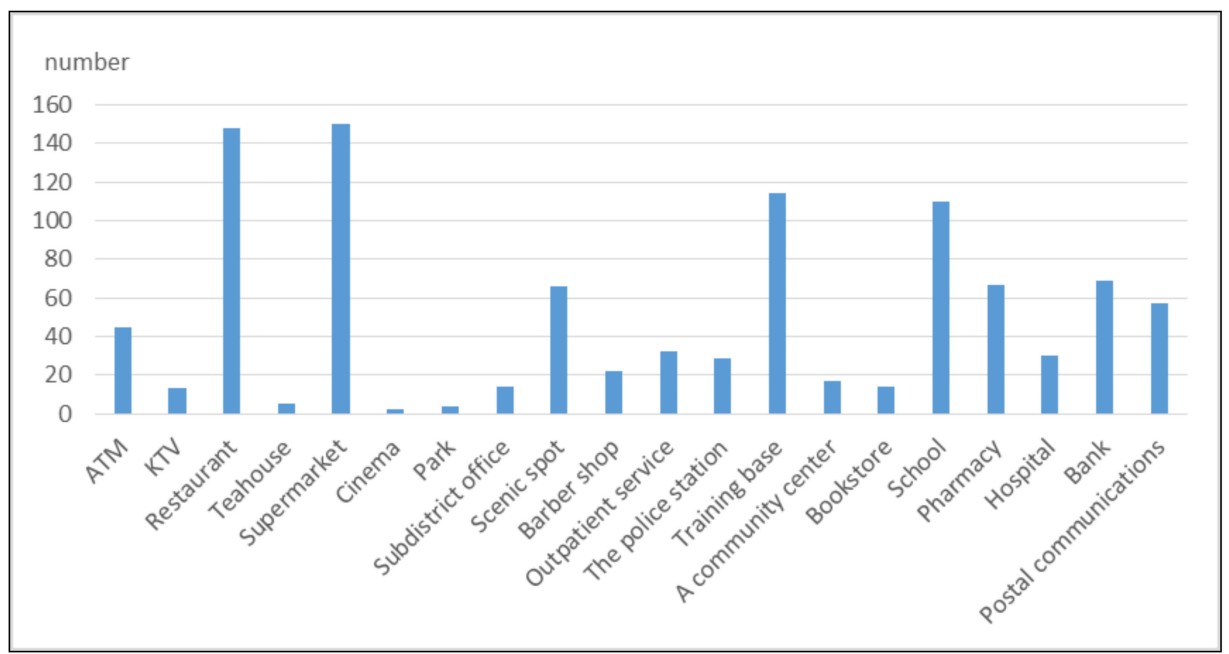

**Figure 9.** Distribution map of functional elements of downtown streets in Gaoping.

### 5.2.2. Visual Construction Elements

The semantic segmentation of street view images is used to obtain the indicators of visual elements, as shown in Figure 10 As seen from Figure 10a, the areas with the open sky are all in the outer areas of the city, and the sky of pedestrian streets around the ancient city area is less open. As seen from Figure 10b, the main pedestrian streets with good green

vision rates are Jianshe road, Hyun street, Kangle street, and residential areas in the north of the city with good greening. As seen from Figure 10c, the areas with relatively high buildings are mainly concentrated in the branch areas. The traditional residential areas with narrow pedestrian streets and lack of shade from green plants make the buildings occupy a relatively high area in the view, giving people a strong sense of oppression. As seen from Figure 10d, the roads with a high proportion of motor vehicle lanes are the roads dominated by motor vehicles, mainly in the newly built roads with wide motor vehicle lanes and some areas that are less founctional.

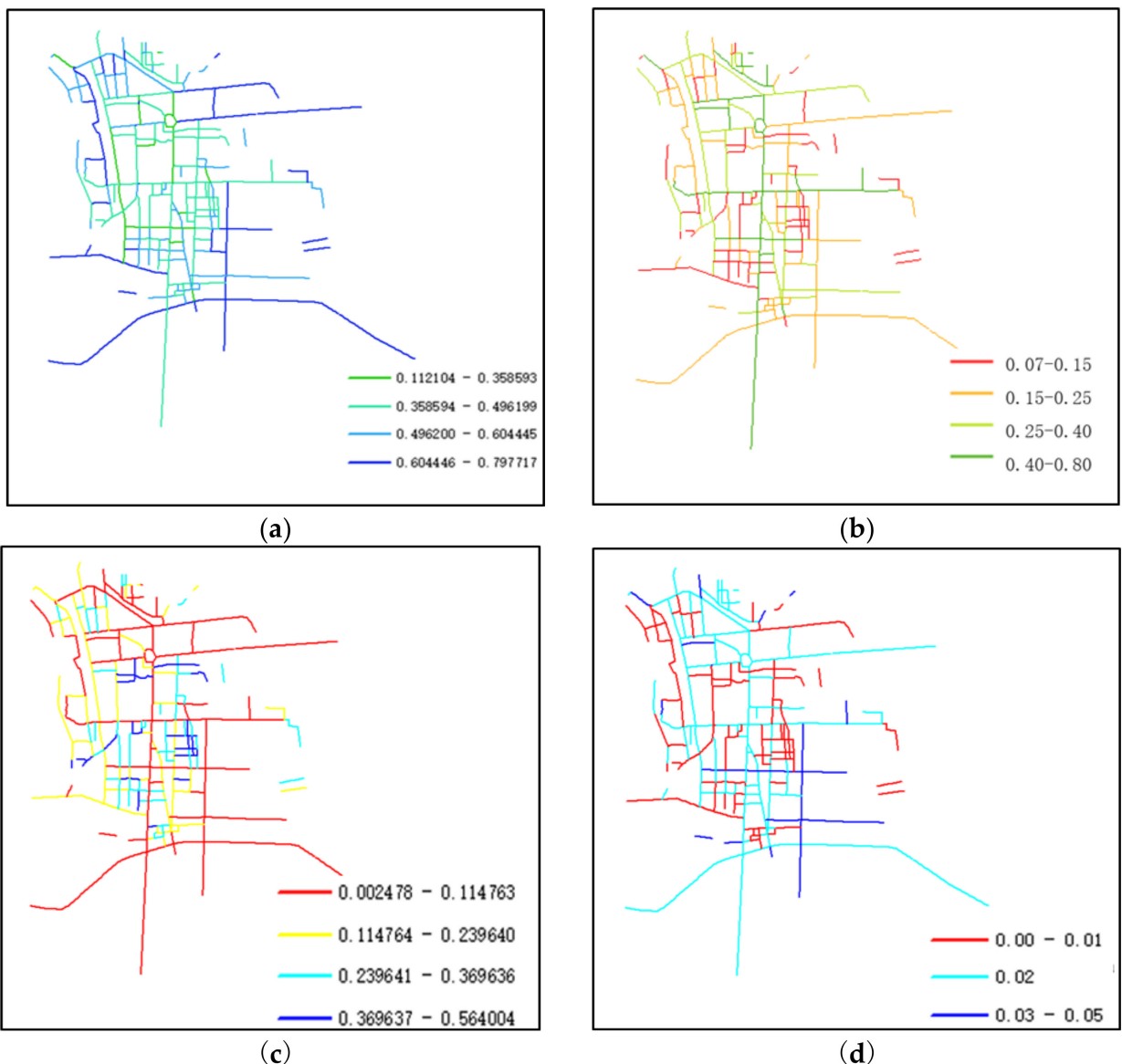

**Figure 10.** Layout of visual elements (**a**) Distribution of sky openness; (**b**) Green rate distribution of the visual field; (**c**) Building ratio distribution map; (**d**) Layout of vehicle lanes.

### 5.2.3. Walking Scale Elements

The width of pedestrian streets in the downtown area of Gaoping was investigated, as shown in Table 6, which includes the width of both sides of the main pedestrian streets. The height and width of streets were calculated, as shown in Figure 11.

**Table 6.** Width of the main pedestrian streets in downtown Gaoping.

| Road Name | Origin | Destination | The Pavement Width (m) | |
|---|---|---|---|---|
| Changping East Street | Square | Danhe North Road | 13 | 6.5 |
| Changpingyuan Road | Changping East Street | Youyi East Road | 1.75 | 1.75 |
| Youyi East Road | Jianshe | Shennong | 10 | 6.5 |
| Shennong North Road | Beiwaihuan | Youyi | 4.5 | 4.5 |
| Changping East Street | Shennong | Shiji | 10 | 8 |
| Qingquan Road | Youyi | Taihua | 3 | 4 |
| Dinglin Road | Youyi | Taihua | 1.5 | 1.5 |
| Taihua West Road | Railway Bridge | Qingquan | 2 | 2.5 |
| Xinjian North Road | Taihua | The Railway Station | 3.5 | 3.5 |
| Yuhong Street | Danhe North Road | Danhe (Bridge) | 3 | 2 |
| Xuanshi East Street | Jianshe Road | Danhe (Bridge) | 6.5 | 4.5 |
| Kangle East Street | Danhe Road | Shennong | 6 | 6 |
| Jingwei Road | Jianghua | Kangle | 5 | 5 |
| Jinghua Street | Taiuo Road | Shennong | 2.5 | 0.5 |
| Gucheng Road | Nanduan | Xuanshi West Street | 2 | 1 |
| Jinfeng East Road | Kangle | Jinghua | 2.5 | 2.5 |
| Xinjian South Road | Jinfeng West Road | Jinghua | 0.75 | 0.75 |
| Jianshe North Road | Youyi Street | Square | 5.5 | 5 |
| Jianshe South Road | Xuanshi Street | Kangle Street | 10 | 12.6 |
| Tailuo Road | Jianshe Road | Shennong Road | 7 | 7 |
| Jianshe South Road | Xinhua Street | Nanwaihuan | 5 | 5 |
| Danhe South Road | Kangle Street | Xinhua Street | 4.5 | 4.5 |
| Yuying Street | Jinshe Road | Danhe Road | 4.5 | 4.5 |
| Xuanshi Street | Jingwei Road | Down Through Century Avenue | 1 | 1 |

The overall integration represents the accessibility of roads in the global road network system. Streets with a higher overall integration have a higher attraction to the aggregation of street social functional facilities due to the larger people flow. When the value is greater than 1, the agglomeration effect of street elements is stronger. When the integration is between 0.4 and 0.6, the layout is more scattered. In the study of the pedestrian street network in the middle and small-medium cities, the overall integration degree represents the pedestrian agglomeration effect of one street in the axis space relative to other street axis space. It reflects the relative advantages of the street axis in the overall street spatial network in terms of accessibility and penetration interaction.

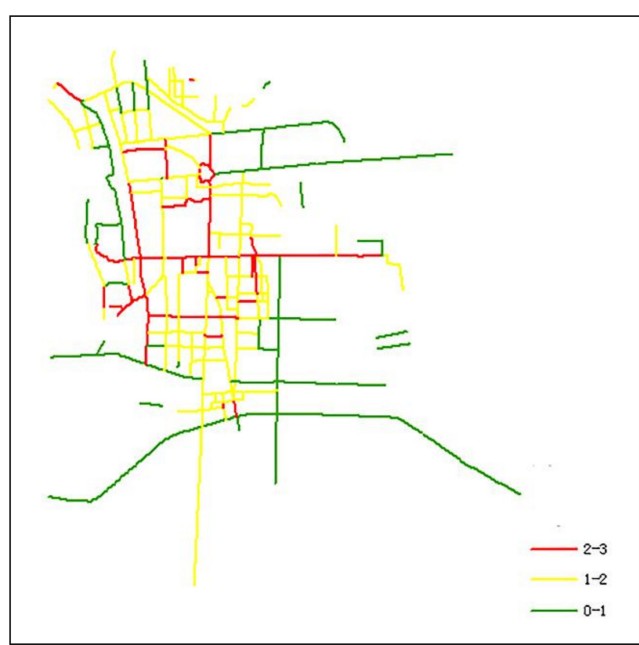

**Figure 11.** Gaoping's road width ratio layout.

## 6. Results and Discussion

According to the street function scores calculated based on the data obtained, pedestrian paths were classified. The classification and scores of the main pedestrian streets are shown in Table 7.

**Table 7.** Classification and scores of main pedestrian streets in Gaoping's downtown.

| Number | Road Number | Pedestrian Path Type | Street Function Score | Visual Perception Score | Walking Scale Score | Total Spatial Environment Score |
|---|---|---|---|---|---|---|
| 1 | Yonghua East Road | Scale bias type | 24 | 12 | 20 | 56 |
| 2 | Yingbin East Road | Scale bias type | 28 | 9 | 21 | 58 |
| 3 | Nanda Street | Scale bias type | 24 | 8 | 17 | 49 |
| 4 | Yingbin Road | Functional bias type | 30 | 5 | 9 | 44 |
| 5 | Jingwei Road | Functional bias type | 32 | 9 | 12 | 53 |
| 6 | Jinghua Street | Visual bias type | 20 | 9 | 12 | 41 |
| 7 | Jianshe Road | Visual bias type | 16 | 13 | 9 | 38 |
| 8 | Youyi Street | Environmental equilibrium bias type | 38 | 11 | 17 | 66 |
| 9 | Xuanshi Street | Functional bias type | 35 | 13 | 9 | 57 |
| 10 | Kangle Street | Visual bias type | 19 | 10 | 12 | 41 |
| 11 | Taihua Road | Visual bias type | 20 | 13 | 11 | 44 |
| 12 | Jianshe Road | Environmental equilibrium bias type | 45 | 11 | 24 | 80 |
| 13 | Changping Street | Functional bias type | 36 | 17 | 23 | 75 |
| 14 | Shennong Road | Functional bias type | 35 | 15 | 17 | 66 |
| 15 | Yandi Avenue | Scale bias type | 20 | 6 | 20 | 46 |
| 16 | Xinhua Street | Visual bias type | 21 | 7 | 9 | 37 |
| 17 | Fu East Road | Visual bias type | 18 | 15 | 11 | 43 |
| 18 | Gucheng Road | Functional bias type | 21 | 11 | 14 | 45 |
| 19 | Jinfeng East Road | Functional bias type | 27 | 5 | 15 | 47 |



### 6.1. Function-Oriented Pedestrian Path Design

According to their scores, Yingbin Road, Jingwei Road, Xuanshi Street, Shennong Road, and Gucheng Road belong to the functional bias type of pedestrian paths. The roads and streets in the ancient city function well, but due to space limitations, there are some problems such as insufficient greening, narrow roads, and low overall integration. To solve these problems, on the one hand, the pedestrian scale can be strengthened to emphasize the advantages of street functions. Similar conclusions have been verified in the city of Cachoeira do Sul [61]. On the other hand, in the existing fish-bone walking path, the connection between the branch road layout and the guillotine can be strengthened, and green plants can be placed in the limited space to reduce the sense of emptiness produced by the open sky.

### 6.2. Visually Oriented Pedestrian Path Design

According to their scores, Jinhua Street, Xinjian Road, Kangle Street, Taihua Road, Xinhua Street, and Fu East Road are visual bias types of pedestrian paths. In the case of Kangle Street, for example, tall trees give walkers a sense of space and appropriate plants make space look less monotonous. The functional diversity of roads should be strengthened to make them more convenient for nearby residents to walk. Researchers have studied the pedestrian environment of smaller Chinese cities (Yiwu and Jinhua), and results revealed similar conclusions [62,63]. The unbalanced street width-to-height ratio can be adjusted by adding street pieces.

### 6.3. Scale Deviation Pedestrian Path Design

According to their scores, Yonghua East Road, Nanda Street, Yingbin East Road, Yandi Avenue, and other pedestrian paths belong to the scale bias type of pedestrian path. As an integrated core axis, Yandi Avenue connects the high-speed rail area with the ancient urban area and has a good pedestrian scale. However, due to the lack of street functions, not many pedestrians use Yandi Avenue. In addition, the open sky and wide motorway create a strong sense of spatial alienation. The functional properties of both sides of the street could be improved by planting taller trees, demolishing old houses appropriately, and constructing a reasonable skyline with new houses. Yu et al. conducted a case study in Shanghai and concluded that street greening improves the sense of spatial alienation [64].

According to the road network score, it can be found that Jianshe Road, Changping Street, Shennong Road, Youyi Street and other roads have higher scores (of more than 60 points), and generally belong to the roads with good road environmental conditions. At the same time, combined with the thermal distribution of travel demand on the Baidu map (shown in Figure 12), it can be confirmed that the areas where residents' travel demand is more concentrated are exactly matched with roads with a road environment score. This means that the study's scoring system can reflect the real conditions of the road.

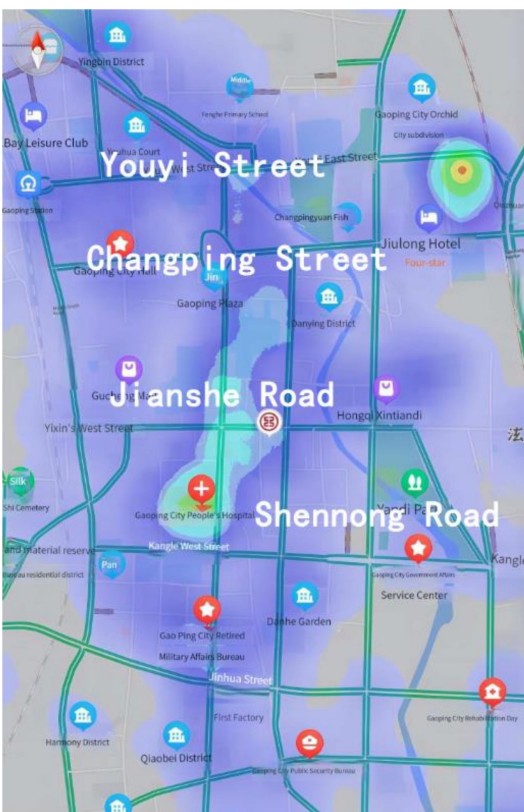

**Figure 12.** The thermal distribution of travel demand on the Baidu map.

## 7. Conclusions

This study examined urban pedestrian traffic planning and design methods for small and medium-sized cities of China based on street walkability. The common way to evaluate Street Walkability is to analyze the functional complexity of the street as a measure of pedestrians' preference for street selection through the POI of the street, and carry out superposition analysis on other factors as a single condition. In this paper, a comprehensive framework of street walkability is developed to measure the influencing factors of pedestrians' choice of walking streets in small and medium-sized cities from multiple dimensions such as street function, visual construction, and walking scale, and to comprehensively evaluate street walkability. Walking distances in small and medium-sized cities and the attenuation coefficient of reasonable facility distance were adopted to modify walking scores. According to the scores obtained, walking paths can be divided into four categories: functionally preferred, visually preferred, scale preferred, and environmentally balanced. Therefore, this study proposes the following policy suggestions for streets and urban spaces in small and medium-sized cities. First, transportation planning should consider the four typical street characteristics mentioned in the article and create a more diverse street environment for the needs of various people. Second, In addition to considering the walkability characteristics of small and medium-sized cities, planning and design departments also need to formulate specific environmental governance technology guidelines when carrying out street design. Third, the evaluation of the pedestrian environment plays an important role in building a green city and promoting residents' physical and mental health to build a healthy city. This categorization provides theoretical support for the design of pedestrian street space environments.

In this study, a limited set of elements of the pedestrian street space environment were considered. In future research, the types of specific indicators considered can be expanded within the existing indicator system. Questionnaires and other more objective models could be used for this purpose, so that the final walking evaluation can guide the design of the walking street environment more objectively. Weights for additional indicators can

be assigned regarding the weights used in the existing model and adjusted as needed to reflect the impact of each element on pedestrians.

Because of the limitations associated with data acquisition, this study focused on a limited set of small and medium-sized cities. Future studies should further analyze the features of urban development and walking in additional small and medium-sized cities. Future studies should also further explore the similarities and differences in pedestrian characteristics between medium-sized and small cities and the reasons for these differences. These would be of great significance in refining the planning and design of pedestrian traffic in small and medium-sized cities. Furthermore, although data on street elements can be acquired quickly using street view pictures, the street landscape varies with seasonal and weather changes, as well as the time of day at which street view pictures are obtained, all of which makes it difficult to describe a street environment objectively. In future studies, it may be useful to employ a time machine function to obtain street view images at different periods for comprehensive analysis which would make it possible to analyze street environments more quantitatively and objectively. Gaps in the ability of street view images can be addressed using handheld devices or by taking photographs from a car.

In this research, the factors and assessment criteria adopted in the assessment of walking environments in small and medium-sized cities are adjusted and modified according to the walking characteristics and residents' travel habits of small and medium-sized cities in China, without taking into account the walking characteristics and customs of small and medium-sized cities in foreign countries. In subsequent studies, under the framework of evaluation, we can localize the evaluation of local indicators and weights according to the travel characteristics and habits of each city for planning and construction guidance. For example, Arellana et al. [65] have introduced features such as road barriers, safety facilities and pedestrian environment into street evaluations in Latin America. The research proves that the street environment has a correlation effect on the choice and guidance of pedestrian travel, and also increases the consideration of road safety and pedestrian's way of thinking. In the future, relevant studies can be continuously improved.

**Author Contributions:** Conceptualization, Y.Z. (Yibang Zhang), X.G. and Z.Z.; methodology, Y.Z. (Yibang Zhang) and Y.Z. (Yukun Zou); software, X.F.; validation, Y.Z. (Yibang Zhang), Y.Z. (Yukun Zou) and Z.Z.; data curation, Z.Z. and X.F.; writing—original draft preparation, Y.Z. (Yibang Zhang) and X.G.; funding acquisition, X.G. All authors have read and agreed to the published version of the manuscript.

**Funding:** This work was supported by the Scientific Research Foundation for Advanced Talents, funder: Nanjing Forestry University, funding number: No.163106041; the General Program of Natural Science Foundation, funder: the Jiangsu Higher Education Institutions of China, funding number: 20KJB580013.

**Institutional Review Board Statement:** Not applicable.

**Informed Consent Statement:** Not applicable.

**Data Availability Statement:** Not application.

**Acknowledgments:** Authors would like to acknowledge the anonymous reviewers for their constructive comments.

**Conflicts of Interest:** The authors declare that they have no conflict of interest.

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
