# Peer review of "Evaluating Pedestrian Environment Using DeepLab Models Based on Street Walkability in Small and Medium-Sized Cities: Case Study in Gaoping, China"

_sustainability, doi:10.3390/su142215472_

Round 1

Reviewer 1 Report

The authors have taken a good initiative to examine urban pedestrian traffic planning and design methods for small and medium-sized cities of China based on street walkability. However, following issues need to be addressed in the revised version if the paper is accepted for publication.

(1)    The main objective of the study is to evaluate pedestrian environment from street walkability perspective for small and medium sized cities. However, in the paper, while assessing street functional score, visual perception score and walking scale score, what would be the possible differences between big cities and small cities from pedestrian point of view are not very clearly mentioned. Why are the findings of all those aspects of big cities not applicable for small and medium cities? More explanation is required in this regard.

(2)    In Table 1, reference is required while presenting the average walking speed for walkers of different age groups.

(3)    As the authors claim that the major contribution of the study is to evaluate the environmental condition which is pedestrian friendly for small and medium sized cities, however, while explaining the results of the study, those are not compared with the findings of big cities. Showcasing differences and similarities the results between big cities and small cities will more clarify the contribution of this research. The results of other countries’ small and medium cities were not compared with the findings of this study.

(4)    In methodology section, many assumptions are formulated without proper citation, however, it seems that those were developed based in line with other studies. Formulating street aspect ratio and walking scale weight are two examples among them.

(5)    Generalizing the findings of this study for all small and medium sized cities based on only one case study would be questionable. Validate the results with the studies of other similar cities elsewhere. This limitation should be discussed in the paper.

(6)    The scale of the pedestrian street environment score was developed in the study. How were the different scales of the value S (pedestrian street environment score) formulated not mentioned in the paper shown in Table 2? Reference literature is required to be sighted in this regard.

(7)    Results and discussion should be more robust and similarities and contrast of the results should be compared with the findings of other similar studies. More critical analysis of the results is required.

(

Author Response

  1. The reviewer 1’s comment:

The main objective of the study is to evaluate pedestrian environment from street walkability perspective for small and medium sized cities. However, in the paper, while assessing street functional score, visual perception score and walking scale score, what would be the possible differences between big cities and small cities from pedestrian point of view are not very clearly mentioned. Why are the findings of all those aspects of big cities not applicable for small and medium cities? More explanation is required in this regard;

Our response:

We agree with the reviewer and thank the reviewer for this constructive comment. In fact, from the perspective of pedestrians, there are similarities and differences between big cities and small cities. The similarity lies in the influence of walking environment on pedestrians and the intervention mechanism. The difference lies in the impact of the urban development scale and characteristics on travel characteristics of pedestrians, as well as the difference in supporting facilities, so the attraction of pedestrian travel is different.

Therefore, this study compares the walking characteristics and walking scale of big cities and small and medium-sized cities, proposes a walking attenuation function suitable for the walking evaluation of small and medium-sized cities, and considers the difference of supporting facilities in small and medium-sized cities, proposes street function elements and corresponding weights in small and medium-sized cities, to modify the overall evaluation model. However, this study lacked an explanation of the difference and comparison between small and medium-sized cities and big cities, so we have added the related elaboration, which is included in the subsequent sections as follows:

In this study, based on the attenuation coefficient setting method in the report Evaluation of Walking Friendliness in Chinese Cities --Research on Street Function to Promote Walking, comparing the characteristics and scale of walking with those of large cities, the walking attenuation coefficient is optimized by combining with the features of small and medium-sized cities.

Based on the 20 classes, street functional elements were divided into 7 categories and 22 subcategories. According to the commonly used supporting functional facilities and travel purposes in the travel characteristics of residents in small and medium-sized cities, this study combines the characteristics and classification of small and medium-sized cities’ street facilities.

  1. The reviewer 1’s comment:

In Table 1, reference is required while presenting the average walking speed for walkers of different age groups;

Our response:

We agree with the reviewer and thank the reviewer for this constructive comment. We have added the new reference.

  1. Pinna, F.; Murrau, R. Age factor and pedestrian speed on sidewalks. Sustainability. 2018, 10, 11, 4084. DOI:10.3390/su10114084

  1. The reviewer 1’s comment:

As the authors claim that the major contribution of the study is to evaluate the environmental condition which is pedestrian friendly for small and medium sized cities, however, while explaining the results of the study, those are not compared with the findings of big cities. Showcasing differences and similarities the results between big cities and small cities will more clarify the contribution of this research. The results of other countries’ small and medium cities were not compared with the findings of this study;

Our response:

We agree with the reviewer and thank the reviewer for this constructive comment. The level of public transport development in small and medium-sized cities is lower than that of large cities. Residents are more dependent on walking. Improving the pedestrian environment in small and medium-sized cities can promote a healthy and civilized lifestyle. In the updated manuscript, we have added comparisons with different types of cities, showing differences and similarities in results between large and small cities. We hope that the contribution of this study is clearly elaborated.

6.1 Function-oriented pedestrian path design

According to their scores, Yingbin Road, Jingwei Road, Xuanshi Street, Shennong Road, and Gucheng Road belong to the functional bias type of pedestrian paths. The roads and streets in the ancient city function well, but due to space limitations, there are some problems such as insufficient greening, narrow roads, and low overall integration. Given these problems, on the one hand, can strengthen the pedestrian scale to emphasize the advantages of street functions. Similar conclusions have been verified in city of Cachoeira do Sul[61]. On the other hand, in the existing fish-bone walking path, the connection between the branch road layout and guillotine can be strengthened, and green plants can be dotted in the limited space to reduce the sense of emptiness produced by the open sky.

6.2 Visually oriented pedestrian path design

According to their scores, Jinhua Street, Xinjian Road, Kangle Street, Taihua Road, Xinhua Street, and Fu East Road are visual bias types of pedestrian paths. In the case of Kangle Street, for example, tall trees give walkers a sense of space and appropriate plants make space look less monotonous. The functional diversity of roads should be strengthened to them more convenient for nearby residents to walk. Researchers have studied the pedestrian environment of smaller Chinese cities (Yiwu and Jinhua), and results revealed similar conclusions[62,63]. The unbalanced street width-to-height ratio can be adjusted by adding street pieces.

6.3 Scale deviation pedestrian path design

According to their scores, Yonghua East Road, Nanda Street, Yingbin East Road, Yandi Avenue, and other pedestrian paths belong to the scale bias type of pedestrian paths. As an integrated core axis, Yandi Avenue connects the high-speed rail area with the ancient urban area and has a good pedestrian scale. However, due to the lack of street functions, not many pedestrians use Yandi Avenue. In addition, the open sky and wide motorway create a strong sense of spatial alienation. The functional properties of both sides of the street could be improved by planting taller trees, demolishing old houses appropriately, and constructing a reasonable skyline with new houses. Yu et al. conducted a case study in Shanghai, and concluded that street greening improves the sense of spatial alienation [64].

  1. Ruiz-Padillo, A.; Oestreich, L.; Torres, T.B.; Rhoden, P.S.; Larranaga A.M.; Cybis, H.B. Weighted assessment of barriers to walking in small cities: A Brazilian case. Transportation Research Part DTransport and Environment. 2022109,103392. DOI:10.1016/j.trd.2022.103392
  2. Yu, Y.C.; Zhao, X.; Zhou, Z.; Zhang, S.; Zhai, D.; Li, J. The associations of built environment with older people recreational walking and physical activity in a Chinese small-scale city of Yiwu. International Journal of Environmental Research and Public Health. 2021, 18(5), 2699. DOI: 10.3390/ijerph18052699
  3. Yu, J.; Yang, C.; Zhang, S.; Zhai, D.; Wang; A.; Li, J. The effect of the built environment on older men′s and women′s leisure-time physical activity in the mid-scale city of Jinhua, China. International Journal of Environmental Research and Public Health. 2021, 18(3), 1039. DOI:10.3390/ijerph18031039
  4. Ye, Y.; Zeng, W.; Shen, Q.; Zhang, X.; Lu, Y. The visual quality of streets: A human-centred continuous measurement based on machine learning algorithms and street view images. Environment and Planning B: Urban Analytics and City Science. 2019, 46(8), 1439–1457. DOI: 10.1177/2399808319828734
  5. The reviewer 1’s comment:

In methodology section, many assumptions are formulated without proper citation, however, it seems that those were developed based in line with other studies. Formulating street aspect ratio and walking scale weight are two examples among them;

Our response:

We agree with the reviewer and thank the reviewer for this constructive comment. We have added references in Distance attenuation rate of walking facilities, Walkway Width, Street Aspect Ratio and Walking scale weight.

  1. 55. Walkability Evaluation of Chinese Cities. Natural Resource Defense Council. 2019. Available online: http://www.nrdc.cn/Public/uploads/2019-10-23/5db00acc2661c.pdf
  2. Liu, Y.N.; Yang, D.J.; Timmermans, J.P.; de Vries, B. Analysis of the impact of street-scale built environment design near metro stations on pedestrian and cyclist road segment choice: A stated choice experiment. Journal of Transport Geography. 2020, 82, 102570. DOI:10.1016/j.jtrangeo.2019.102570
  3. Yao, G.; Yuan, T.T.; Rui. Y.; Chen, W.J.; Duan, Z.C.; Sun, L.; Si, X.Z.; Zhang, M.; Chen, K.Y.; Zhu, Y.S.; Chen; Y.Y. Research on the scale of pedestrian space in underground shopping streets based on VR experiment. Journal of Asian Architecture and Building Engineering. 2021, 20(2), 138-153. DOI:10.1080/13467581.2020.1782215
  4. Tang, J.; Long, Y. Measuring visual quality of street space and its temporal variation: Methodology and its application in the Hutong area in Beijing. Landscape and Urban Planning. 2018, 191, 103436. DOI:10.1016/j.landurbplan.2018.09.015
  5. Li, S.J.; Ma, S.; Tong, D.; Jia, Z.M.; Li, P.; Long Y. Associations between the quality of street space and the attributes of the built environment using large volumes of street view pictures. Environment and Planning B: Urban Analytics and City Science. 2021, 4(49), 1197-1211. DOI:10.1177/23998083211056341

  1. The reviewer 1’s comment:

Generalizing the findings of this study for all small and medium sized cities based on only one case study would be questionable. Validate the results with the studies of other similar cities elsewhere. This limitation should be discussed in the paper;

Our response:

We agree with the reviewer and thank the reviewer for this constructive comment. In this study, the impact of walking environment on pedestrians in small and medium-sized cities is considered, and this method is applied to a small city in China for preliminary demonstrations. The evaluation system and evaluation method proposed in this study are practiced and applied, which has been preliminary proved feasible. However, as mentioned that the discussion regarding the limitations of the application of the method is still lacking. On the one hand, due to the influence of space, the discussion is not comprehensive; on the other hand, due to the numerous types and wide scope of small and medium-sized cities, it cannot be discussed one by one, so it needs to be further improved in subsequent researches. Therefore, so we have added the related elaboration, which is included in the subsequent sections as follows:

In this study, the factors and assessment criteria adopted in the assessment of walking environment in small and medium-sized cities are adjusted and modified based on the walking characteristics and residents' travel characteristics, without taking into account the walking characteristics of small and medium-sized cities in foreign countries. In the subsequent studies, regarding the framework of the evaluation, each city travel characteristics and travel characteristics should be considered to meet local indexes and weights of assessment requirements for planning and construction guidances. For example, Arellana [65] conducted that streets were evaluated in Latin America introduced the road barriers and safety facilities, and should consider the pedestrian environment.

  1. Arellana, J.; Saltarín, M.; Larrañaga, A.M.; Alvarez, V.; Henao, C.A. Urban walkability considering pedestrians’ perceptions of the built environment: A 10-year review and a case study in a medium-sized city in Latin America. Transport reviews. 2020, 40(2), 183-203. DOI:10.1080/01441647.2019.1703842

  1. The reviewer 1’s comment:

The scale of the pedestrian street environment score was developed in the study. How were the different scales of the value S (pedestrian street environment score) formulated not mentioned in the paper shown in Table 2? Reference literature is required to be sighted in this regard;

Our response:

We agree with the reviewer and thank the reviewer for this constructive comment. Regarding the scoring criteria in Table 2, this study refers to the following literature and makes supplements. We have added references in Table 2. The scale of the pedestrian street environment score.

  1. Zhang, Y.X.; Zhang, J.Z.; Xu, K.; Tang, D.M.; Li, Y.; Wang, X.C.; Zhang, K. An improved method for urban walk score calculation considering perception of the street environment. Transactions in Gis. 2022, 3(26), 1399-1420. DOI:10.1111/tgis.12909

  1. The reviewer 1’s comment:

Results and discussion should be more robust and similarities and contrast of the results should be compared with the findings of other similar studies. More critical analysis of the results is required.

Our response:

We agree with the reviewer and thank the reviewer for this constructive comment. In the discussion of conclusions and limitations, this paper only considers its own research and lacks comparison with other similar studies. In order to increase the credibility of the conclusions and compare with other research results, So we have added described what is included in the subsequent sections as follows:

Arellana [65] conducted Latin America from 2009 to 2018, the research on index on foot, by studying the certificate in accordance with this article conclusion, street environment for pedestrians travel choice and guidance there is a correlation function, and also increase the corresponding road safety and pedestrian way of thinking, more perfect to pedestrians travel consideration, is the future related research can continue to improve.

  1. Arellana, J.; Saltarín, M.; Larrañaga, A.M.; Alvarez, V.; Henao, C.A. Urban walkability considering pedestrians’ perceptions of the built environment: A 10-year review and a case study in a medium-sized city in Latin America. Transport reviews. 2020, 40(2), 183-203. DOI:10.1080/01441647.2019.1703842

Reviewer 2 Report

The paper can be improved based on the following:

The introductory section describes the problem statement well, but ideally, it should be concluded with some reference to the purpose of the paper and an indication of what is included in each of the subsequent sections.

Why haven't travel purposes (described in lines 282-287) related to work trips, or access to services (ie. healthcare) been considered?

What is the difference between facilities (described on page 4) and the 7 categories of functional elements stated on page 9 (lines 309-311)?

It is not very clear how the weights presented in Table 4 have been derived. Some further explanation will be beneficial to the readers.

The figures in section 5.2.2 are not legible.  Also, how were the Visual Construction Elements analysed. Were different photographs used? How many? Some further explanations will help the reader.

The rationale for the generation of the results presented in section 6 is sound, but a validation/verification of the results is not present. How can the authors verify that the scores reflect the actual behaviour of pedestrians?

Reviewer 3 Report

This is an interesting study in the area of the active transport. I recommend you add some photos from the studied streets to better show case the differences between the provided scores in Table 7. Photos should compares streets from street function, visual perception, walking score etc.  

Round 2

Reviewer 1 Report

The revised paper can be accepted now.

Reviewer 2 Report

The authors have addressed the comments raised in the previous round of reviews and therefore the paper can be accepted in its current form.